# MoS_2_ as a Co-Catalyst for Photocatalytic Hydrogen Production: A Mini Review

**DOI:** 10.3390/molecules27103289

**Published:** 2022-05-20

**Authors:** Sayyar Ali Shah, Iltaf Khan, Aihua Yuan

**Affiliations:** School of Environmental & Chemical Engineering, Jiangsu University of Science and Technology, Zhenjiang 212100, China; sayyar786@gmail.com

**Keywords:** photocatalysis, heterojunction, layers structure materials, hydrogen production

## Abstract

Molybdenum disulfide (MoS_2_), with a two-dimensional (2D) structure, has attracted huge research interest due to its unique electrical, optical, and physicochemical properties. MoS_2_ has been used as a co-catalyst for the synthesis of novel heterojunction composites with enhanced photocatalytic hydrogen production under solar light irradiation. In this review, we briefly highlight the atomic-scale structure of MoS_2_ nanosheets. The top-down and bottom-up synthetic methods of MoS_2_ nanosheets are described. Additionally, we discuss the formation of MoS_2_ heterostructures with titanium dioxide (TiO_2_), graphitic carbon nitride (g-C_3_N_4_), and other semiconductors and co-catalysts for enhanced photocatalytic hydrogen generation. This review addresses the challenges and future perspectives for enhancing solar hydrogen production performance in heterojunction materials using MoS_2_ as a co-catalyst.

## 1. Introduction

Hydrogen is a clean, renewable energy source and alternative to fossil fuels [1] that can be stored at high mass-specific energy density, and its only product on combustion is water [2]. At present, about 96% of hydrogen is industrially produced from coal gasification and steam methane reformation processes [1,2]. However, these processes of hydrogen production also generate secondary pollutants or greenhouse gases, such as CO_2_ and N_2_O, that affect the environment [2]. Methane pyrolysis produces hydrogen and solid carbon as a byproduct [3]. This process generates CO_2_-free hydrogen and has an advantage over conventional steam methane reformation and coal gasification processes. However, methane pyrolysis is a temporary solution and not a sustainable process due to the depletion of natural gas reserves [3].

To overcome energy challenges and environmental problems, hydrogen production from electrochemical water splitting using highly active catalysts is a promising strategy [4,5]. Less than 4% of hydrogen is produced through electrocatalysis at the industrial level [2]. The electrocatalysis of water for hydrogen production is a high-cost technique, which has hindered its large-scale industrialization. As an alternative, photocatalytic hydrogen evolution reaction (HER) from water splitting over a particular semiconductor material has been the most interesting way to address these issues. Generally, the photocatalytic efficiency depends upon three processes, including light absorption in the solar spectrum, charge separation, and surface active sites for catalytic activity [1,2,6].

A photocatalyst that can absorb sunlight across the whole solar spectrum is considered to be an ideal candidate for photocatalysis [6,7]. In 1972, Fujishima et al. reported photo-induced water splitting on TiO_2_ electrodes [8]. Since then, much research has been focused on TiO_2_ and other related semiconducting materials such as metal oxides, metal sulfides, conjugated polymers, nanosheets, graphitic carbon nitride, metal organic frameworks, and covalent organic frameworks, etc., as photocatalysts for hydrogen production [9,10,11,12,13,14,15,16,17]. However, the available photocatalysts for hydrogen production are still limited due to low visible light absorption and high electron–hole recombination rates.

Molybdenum disulfide (MoS_2_), with a 2D nanostructure, has attracted huge attention due to its outstanding optical and electronic properties and promising applications [18,19,20,21,22]. MoS_2_ nanomaterials as co-catalysts are promising photocatalysts for HER [18,23]. It is reported that the exposed edges of layers of MoS_2_ contain active sites for catalytic activity while its basal planes are mostly inactive [19,24]. In addition to the active photocatalytic sites, the band gap of MoS_2_ nanosheets is an important parameter for photocatalytic HER. The band gap of MoS_2_ increases from bulk (1.2 eV) to single layer (1.9 eV) due to quantum confinement [19,24]. As a result, the location of the conduction band (CB) of MoS_2_ moves towards a more negative potential than the proton reduction potential (H^+^/H_2_), which consequently enhances the reduction in adsorbed H^+^ and photocatalytic hydrogen evolution.

It is widely reported that loading a co-catalyst over semiconductors is a promising approach with superior photocatalytic performance due to the photoelectron separation and charge transfer [18,19]. MoS_2_-decorated semiconductor materials constitute a promising approach that has shown superior hydrogen production due to their heterojunctions with controllable nanoscale architectures, design for enhanced performance in terms of light absorption, charge separation, and surface catalytic reactions [15,19,23,24].

In this review, we briefly introduce the basic aspects and synthetic methods of MoS_2_ nanosheets. Different types of MoS_2_-based heterojunction composites are also discussed. The role of MoS_2_ nanomaterials as co-catalysts in heterojunction composites for enhanced HER performance is addressed. Additionally, some important issues are highlighted and useful opinions are discussed to further improve photocatalytic hydrogen production using MoS_2_ as a co-catalyst.

## 2. Atomic-Scale Structure of MoS_2_

A single layer of MoS_2_ has a sandwich structure of S-Mo-S, where the Mo atoms are covalently bonded with the S atoms (Figure 1). MoS_2_ has several polymorphs, including 1T_1_, 1T_2_, 1H, 2T, 2H, 2R_1_, 2R_2_, 3H_a_, 3H_b_, 3R, and 4T [25,26,27,28,29]. Among them the 1T MoS_2_, 2H MoS_2_, and 3R-MoS_2_ polymorphs of MoS_2_ have been most investigated for different applications [25,27,28,29]. A single-layer 1T MoS_2_ sheet is metallic and has good electrical properties [30,31], while single-layer 2H MoS_2_ and 3R-MoS_2_ sheets behave as a semiconductor with a direct band gap [28,32].

Generally, MoS_2_ sheets are stacked together by weak van der Waals forces and form few-layer MoS_2_ nanosheets. As the band gap of MoS_2_ nanosheets increases from bulk (1.2 eV) to single layer (1.9 eV) [33], it absorbs the visible region of the solar spectrum. Thus, MoS_2_ can play an important role as a co-catalyst during photocatalysis [29]. MoS_2_-based semiconductor composites act as co-catalysts that can significantly enhance the efficiency of photocatalytic hydrogen production [7,34,35,36,37].

## 3. Photochemical Hydrogen Evolution Reaction

As mentioned above, Fujishima and Honda reported on photo-induced water splitting on TiO_2_ electrodes. Hydrogen can also be directly produced from photochemical water splitting. Usually, a photoelectrolytic cell is designed to carry out the photochemical water splitting process. A typical photoelectrolytic cell for water splitting is shown in Figure 2a [38]. Using light sources, the photocatalytic water splitting takes place in several steps: the absorption of light by catalyst on electrode; the generation of charges followed by the excitation of electrons in the valence band; the separation of charge as well as the transport of charge carriers; and the oxidation of water and generation of hydrogen during water splitting, which occur at separate electrodes. The pure, overall water splitting process comprises two half-reactions to generate hydrogen and oxygen molecules, as shown in Figure 2b [39]. Water oxidation occurs at the anode to produce oxygen, whereas H^+^ ions are reduced on the cathode into hydrogen gas. For more details of photocatalytic water splitting, see the review of Jeong et al. [39].

## 4. Synthesis of MoS_2_

Nanostructured MoS_2_ can be fabricated via both top-down and bottom-up approaches. In the case of the top-down method, the commercially available bulk crystal of MoS_2_ is physically downsized into MoS_2_ nanomaterials (Figure 3) [29,40,41], while in the bottom-up approach, MoS_2_ nanomaterials are synthesized via chemical reaction with small molecules using chemical vapor deposition (CVD) and hydrothermal or solvothermal methods, etc. [42,43,44]. Single layers, multilayers, nanoparticles, and quantum dots of MoS_2_ have also been reported [45,46,47,48]. Continued efforts have been reported for the fabrication of MoS_2_ nanomaterials via the top-down and bottom-up strategies [16,17,18,19,28,29,30,31,40,41,42,43,44].

### 4.1. Top-Down Approach

#### Exfoliation of MoS_2_

Due to the layered structure and van der Waals interactions, MoS_2_ nanosheets can be easily prepared through the exfoliation method. Mechanical, chemical, electrochemical, and liquid-phase exfoliation processes have been reported for the synthesis of MoS_2_ nanosheets [39,40,41,42,43,44,45,46,47,48,49,50,51,52]. For example, in the mechanical exfoliation technique, the suitable MoS_2_ flakes are peeled off from the bulk crystal of MoS_2_ using adhesive tape and shifted onto a specific substrate [46,53]. When the scotch tape is detached, some parts of MoS_2_ remain on the substrate. As result, single- or few-layer MoS_2_ nanosheets with random shapes and sizes are obtained. The 2D materials prepared by the exfoliation method have good quality and allow to study the pristine properties of materials and device performance. However, during this process, the thickness and size of the MoS_2_ are difficult to control, and the resulting materials are inappropriate for large-scale production and scaled-up applications [53,54]. Li et al., mechanically exfoliated single- and multilayer MoS_2_ nanosheets from SiO_2_/Si with the adhesive tape method [41]. The flakes of MoS_2_ were mechanically stripped on Si/SiO_2_ substrate. The obtained single-layer and multilayer MoS_2_ materials were characterized using a bright-field optical microscope and an atomic force microscope (AFM). From the AFM measurements, the height of a single MoS_2_ sheet was found to be 0.8 nm, while the thickness of two, three, and four layers of MoS_2_ nanosheets was 1.5, 2.1, and 2.9 nm, respectively (Figure 4). The MoS_2_ nanosheet monolayers showed an enhanced optical performance, especially single-layer MoS_2_ nanosheets. It was observed that the van der Waals interactions between MoS_2_ to SiO_2_ were much weaker. For this purpose, gold can be used as a substrate to exfoliate the MoS_2_ nanosheets due to its strong affinity for sulfur. It can exfoliate the MoS_2_ monolayer from the bulk because of the strong van der Waals interactions between Au and MoS_2_ layers [55,56,57]. Huang et al. prepared large-area MoS_2_ nanosheets using a Au-assisted exfoliation strategy [50]. In a typical synthesis, a Au thin layer was deposited on a Ti or Cr adhesion-covered substrate. To develop good contact between a MoS_2_ bulk crystal on tape and a Au-covered substrate, it should be passed under high pressure. The monolayer sheets with a large area were collected from the surface of the Au after peeling off the tape.

In the top-down approaches, single- and multilayer MoS_2_ nanosheets are prepared, which have been used to study some fundamental properties of MoS_2_ nanosheets.

### 4.2. Bottom-Down Approach

#### 4.2.1. Chemical Vapor Deposition

The CVD technique has a long history and is commonly used for the synthesis of high-quality semiconductor materials. In a typical CVD process of MoS_2_ nanosheets, the Mo sources are solid precursors of Mo or MoO_3_ powder, and the S sources are H_2_S gas or solid S powder [58,59,60,61]. The solid MoO_3_ and vaporized S react with each other in a low-pressure chamber, forming nuclei for the growth of MoS_2_ [58]. Then, MoS_2_ slowly grows and enlarges its size on the substrates under carrier gas flow. The temperatures at which MoS_2_ grows during the CVD process are usually between 700 and 1000 °C, with a metal catalyst such as Au [61]. Plasma-enhanced CVD requires a low temperature (150–300 °C) for the growth of MoS_2_ nanosheets, and MoS_2_ can even be directly deposited on the plastic substrate [62]. Recently, metal organic CVD has been reported for the synthesis of MoS_2_ nanosheets [63,64], where organometallic precursors were used as starting materials.

#### 4.2.2. Physical Vapor Deposition

Advanced technology such as molecular beam epitaxy (MBE) can be used to prepare single-crystal semiconductor thin films. However, its applications are limited to the synthesis of 2D materials [65]. Ordinary physical vapor deposition is rarely reported for 2D materials. A MoS_2_–Ti composite was prepared by direct current magnetron sputtering, using Ti and MoS_2_ materials [66]. In this process, the MoS_2_ was amorphous.

#### 4.2.3. Solution-Based Process

Solution-based processes are commonly used to synthesize MoS_2_ nanosheets. Hydrothermal and solvothermal methods are the most interesting for the preparation of MoS_2_ nanosheets [67,68]. In these methods, the Mo source is commonly a molybdate, such as Na_2_MoO_4_ or (NH_4_)_6_Mo_7_O_24_, and the S source is thiourea and thioacetamide and L-cysteine [69,70,71,72,73]. The molybdate reacts with the S or S compound in a stainless steel autoclave. The physicochemical reaction takes place at high temperatures (160–200 °C) and pressure for at least a few hours. In the solvothermal method, organic solvents such as 1-methyl-2-pyrrolidinone, N,N-dimethylformamide, and polyethylene glycol-600 are used to proceed with the reaction, while in the hydrothermal method, water is used as a solvent. The MoS_2_ powders obtained from these methods have different sizes and shapes. The sizes and shapes of the products can be adjusted by altering the experimental conditions. To improve the crystalline quality of MoS_2_, the products are usually post-annealed at high temperature.

The MoS_2_ nanomaterials prepared through different bottom-up approaches have various sizes, shapes, morphologies, and thicknesses and can be used for many applications.

## 5. Application of MoS_2_ as a Co-Catalyst in Photocatalysis for Hydrogen Production

### 5.1. MoS_2_/Titanium Dioxide Composites

The semiconducting material titanium dioxide (TiO_2_) has been employed for hydrogen production due to its good UV light response, non-toxic nature, low cost, chemical stability, and good availability [1,12,24]. However, the photocatalytic energy conversion efficiency of TiO_2_ for hydrogen production is low due to its wide band gap structure (Eg ≈ 3.2 eV), photogenerated charge recombination, and some reverse reactions [1,2,12,24]. Many strategies have been attempted to improve the catalytic activity of TiO_2_ nanomaterials, including micro/nanostructure constructing, crystal facet, crystal phase, surface, and tailoring the band gap [9,74,75], but the photocatalytic activity of TiO_2_ still cannot reach the expected efficiency.

MoS_2_ is considered a potential co-catalyst for TiO_2_ materials to boost the efficiency of photocatalytic hydrogen production. Zhu and his coworkers fabricated MoS_2_/TiO_2_ photocatalysts with various compositions through a facial mechanochemistry method [76]. The photocatalytic activity of the prepared composite was studied for hydrogen generation under UV irradiation. The 4% MoS_2_ loaded on TiO_2_ (4%-MoS_2_/TiO_2_) showed maximum hydrogen production at a rate of 150.7 μmol h^−1^, which is about 48.6 times higher than that of pure TiO_2_ at ~3.1 μmol·h^−1^. The improved photocatalytic activity of MoS_2_/TiO_2_ composites is mainly due to electron transfer from TiO_2_ to MoS_2_ nanosheets and the active sites that produce hydrogen. Meanwhile, the recombination rate of electron–hole pairs is also reduced. Furthermore, the relatively good conductivity of MoS_2_ nanosheets also assisted the photo-induced charge separation, leading to an enhanced photocatalytic performance. Ma et al. reported flower-like MoS_2_/TiO_2_ nanohybrid composite photocatalysts obtained from a metal organic framework-derived precursor via facial hydrothermal methods [77]. The flower-like morphology of the MoS_2_/TiO_2_ composites was confirmed from SEM images, as shown in Figure 5. In order to investigate the photocatalytic activity, the experiments were conducted under visible light conditions with fluorescein as a photosensitizer. An outstanding improvement in the photocatalytic activity was achieved for the optimized sample (14.6 wt% MoS_2_ loaded on TiO_2_) with a hydrogen evolution rate of 10046 μmol·h^−1^·g^−1^. They concluded that this high performance of the MoS_2_/TiO_2_ composites is associated with the formation of active centers as well as the uniform distribution of MoS_2_ and TiO_2_ phases, inducing electrons’ motion to reduce protons. In the proposed photocatalytic activity mechanism, excited electrons from fluorescein transfer to the CB of TiO_2_. These electrons further move to the surface of MoS_2_ and combine with protons to produce hydrogen. Liu and his coworker prepared MoS_2_ nanosheets rooted in TiO_2_ nanofibers (TiO_2_@MoS_2_) using a hydrothermal strategy [19]. They reported single- to few-layer MoS_2_ nanosheets and TiO_2_ nanofibers’ porous structure. The MoS_2_ nanosheets grew vertically on the porous structure of TiO_2_, and deep rooting MoS_2_ nanosheets into TiO_2_ nanofibers put them in close contact for the electron transfer process and structural stability. The hydrogen production rates of the TiO_2_@MoS_2_ sample were 1.68 under UV–vis light and 0.49 mmol·h^−1^·g^−1^ under visible light.

TiO_2_ nanomaterials combined with a MoS_2_ co-catalyst can enhance hydrogen production rates up to several times.

### 5.2. MoS_2_/Graphitic Carbon Nitride Composites

Graphitic carbon nitride (g-C_3_N_4_) is considered one of the promising candidates for photocatalysis due to its high chemical stability, environmentally friendly nature, and suitable energy bands that can efficiently absorb solar spectrum irradiation [78,79,80,81]. However, g-C_3_N_4_ suffers from a small specific surface area, high exciton binding energy, stacking back into a bulk, and low efficiency under visible light [82,83,84]. Recently, much interest has been devoted to g-C_3_N_4_-based composites for solar hydrogen production under visible light. To enhance the efficiency of its photocatalytic activity, various non-precious co-catalysts such as Co_2_P, Mo_2_C, and MoS_2_ have been incorporated with C_3_N_4_ [85,86,87]. Among them, MoS_2_ as a co-catalyst in MoS_2_/C_3_N_4_ composites shows promising efficiency for photogenerated hydrogen production [87,88].

The design of and nano-interface coupling between MoS_2_ and C_3_N_4_ can significantly enhance the photocatalytic HER performance. The appropriate MoS_2_/C_3_N_4_ composites with an optimal ratio are believed to enhance solar absorption, increase the interfaces, and decrease the electron transfer distance of the photo-excited electrons between C_3_N_4_ and MoS_2_ co-catalysts. Yuan’s group reported MoS_2_/g-C_3_N_4_ composites with various contents of MoS_2_ developed using the solvent thermal method. The composite catalysts were evaluated for photocatalytic H_2_ generation [87]. They found that MoS_2_/g-C_3_N_4_ composites containing 0.75% MoS_2_ nanosheets performed better and had a reaction rate of 1155 μmol·h^−1^·g^−1^ under visible light irradiation. The apparent quantum yield was about 6.8% under a monochromatic light of 420 nm. Furthermore, they explained that the large surface area of g-C_3_N_4_ nanosheets and the nano-interface coupling between MoS_2_ nanosheets and g-C_3_N_4_ were mainly responsible for the outstanding photocatalytic hydrogen production of the MoS_2_/g-C_3_N_4_ composite. Recently, Li et al., reported the in situ synthesis of a g-CN/MoS_2_ composite [89]. The composite exhibited enhanced photocatalytic hydrogen production compared to pristine g-CN under visible light irradiation. The rod-like MoS_2_ plays an important role as co-catalyst in the g-CN/MoS_2_ composite in the enhancement of the hydrogen production rate. Zhang et al. reported sulfur-doped C_3_N_4_ with covalently crosslinked MoS_2_ nanosheets (MoS_2_/SC_3_N_4_) for improved photocatalytic hydrogen production [88]. The ultrathin array-like nanosheet structure of the MoS_2_/SC_3_N_3_ composites was observed by SEM characterizations (see Figure 6). MoS_2_/SC_3_N_3_ composites were studied for photocatalytic HER under visible light conditions. MoS_2_/SC_3_N_3_ with 2.5% MoS_2_ nanosheets showed the optimal hydrogen production rate of 702.53 μmol·h^−1^·g^−1^. The array-like porous morphology had a rich exposed surface, covalent bonding structure, and enhanced visible light absorption by the cyano group of MoS_2_/SC_3_N_3_ composites. This facilitates the photogenerated electrons’ transfer from the CB of SC_3_N_3_ to MoS_2_ via a heterojunction interface that consequently enhances the photocatalytic hydrogen evolution. Zhang et al. reported a MoS_2_/Fe_2_O_3_/g-C_3_N_4_ ternary composite photocatalyst under hydrothermal conditions for hydrogen production [90]. The obtained ternary composite showed a hydrogen production rate about five times higher compared to g-C_3_N_4_. In addition, 1T MoS_2_/C_3_N_4_ composites also show enhanced photocatalytic hydrogen production [91,92,93,94]. Li et al. loaded metallic 1T-phase MoS_2_ quantum dots onto CdS nanorods (1T-MoS_2_-CdS) using a one-step hydrothermal method at different temperatures [91]. The 1T-MoS_2_-CdS composite prepared at 180 ºC showed remarkable photocatalytic hydrogen production (131.7 mmol·h^−1^·g^−1^) under visible light (λ > 420 nm). This rate of hydrogen evolution reaction was over 65 times greater than that of pure CdS (mmol·h^−1^·g^−1^) and two times that of Pt-loaded CdS.

Besides 1T-phase MoS_2_, amorphous MoSx nanomaterials are efficient electrocatalysts as well as co-catalysts for hydrogen production [95,96,97]. They provide more unsaturated active S atoms, which can rapidly capture protons from the solution to convert them into hydrogen molecules. Yu et al. reported amorphous MoSx/g-C_3_N_4_ (a-MoSx/g-C_3_N_4_) composites developed using an adsorption in situ transformation method [95]. The a-MoSx/g-C_3_N_4_ composites were compared with crystalline MoS_x_/g-C_3_N_4_ and g-C_3_N_4_ catalysts, and all of the a-MoSx/g-C_3_N_4_ catalysts displayed better photocatalytic performances than the crystalline MoSx/g-C_3_N_4_ and C_3_N_4_ catalysts. Among the a-MoSx/g-C_3_N_4_ composites, the a-MoSx/g-C_3_N_4_ catalyst with 3 wt% Mo showed the best photocatalytic performance and a hydrogen production rate of 273.1 μmol·h^−1^·g^−1^.

Similar to TiO_2_/MoS_2_ photocatalysts, MoS_2_/g-C_3_N_4_ heterojunction composites can improve hydrogen production.

### 5.3. MoS_2_ Coupling with Other Semiconductor Materials

As discussed earlier, MoS_2_ as a co-catalyst for other semiconductor compounds can efficiently enhance the photocatalytic activity of hydrogen generation. The interfacial coupling of MoS_2_ with semiconductor compounds has been designed in many strategies. An appropriate ratio, increased interface area, and decreased migration distance of the photogenerated electrons between the MoS_2_ and the semiconductor compounds can effectively improve photocatalytic hydrogen production. Zhang et al. reported MoS_2_/CdS composites with willow branch-shaped morphology developed using a one-pot hydrothermal method [98]. The MoS_2_/CdS composite with 5 wt% MoS_2_ as a co-catalyst displayed an enhanced photocatalytic performance and produced 250.8 μmol·h^−1^ hydrogen evolution with an apparent quantum efficiency of 3.66% at 420 nm. Preparation of the willow branch-shaped nano-heterojunction morphology enhances the visible light absorption and also promotes the separation of photogenerated electron–hole pairs.

Ma et al. reported a layered CdS/MoS_2_ heterostructure photocatalyst developed using ultrasonicated MoS_2_ and CdS nanosheets, produced from hydro- and solvothermal methods, respectively [99]. When MoS_2_ co-catalysts were loaded onto CdS nanosheets, the photocatalytic performance of the CdS/MoS_2_ heterostructure was twice that of the pure CdS photocatalyst. The designing of a layered CdS/MoS_2_ heterostructure could efficiently enhance the photogenerated charge separation and electron transfer, which improves the surface hydrogen evolution kinetics. Patriarchea and coworkers synthesized CdS nanoparticles using polymer-templated oxidative aggregation, and subsequently, MoS_2_ nanosheets were deposited on it via the wet chemical method [100]. The obtained optimized MoS_2_/CdS catalyst showed a good hydrogen production rate of about 0.4 mmol h^−1^ under visible light compared to the CdS catalyst. The enhanced hydrogen generation was due to the presence of the MoS_2_ co-catalyst.

Samaniego-Benitez and coworkers prepared ZnS/MoS_2_ heterostructure materials using a one-pot solvothermal method [101]. The hydrogen production yield of the ZnS/MoS_2_ sample with 10% Mo reached 2600 μmol·h^−1^ under UV light for 4 h. They concluded that the enhanced photocatalytic activity was due to the synergistic effect between ZnS and MoS_2_ and sulfur vacancies created in the ZnS structure during the synthesis process. In the proposed mechanism, a photoexcited electron moves from the CB of ZnS to the CB of MoS_2_, where it interacts with the proton and produces hydrogen.

Recently, Guan et al., used MoS_2_ as a co-catalyst for methylammonium lead iodide to split hydrogen iodide for photocatalytic HER [102]. The methylammonium lead iodide microcrystals and MoS_2_ nanoflowers (MAPbI_3_/MoS_2_) formed a heterostructure. The MoS_2_ nanoflowers have plenty of active catalyst sites for hydrogen evolution. The hydrogen evolution rate of MAPbI_3_/MoS_2_ reached ~30,000 μmol·h^−1^·g^−1^ and a solar-derived hydrogen iodide splitting efficiency of 7.35% was achieved under visible light irradiation. This hydrogen evolution rate is more than 1000 times higher compared to that of pristine MAPbI_3_. The MoS_2_ can induce charge separation and provide abundant active sites for photocatalytic hydrogen evolution.

For these examples, we can conclude that MoS_2_ is an efficient co-catalyst for CdS, ZnS, and MAPbI_3_ etc, catalysts to produce hydrogen.

### 5.4. MoS_2_ and Other Co-Catalyst Heterojunction Composites

The heterojunction of a MoS_2_ co-catalyst with other co-catalysts is an attractive strategy because it can improve the photogenerated electron transfer from a semiconductor to a MoS_2_ co-catalyst during photocatalysis, which enhances the activity via the catalytic sites on MoS_2_ co-catalysts [103,104,105]. The heterojunctions between MoS_2_ and highly conductive co-catalysts decrease the resistance effect and increase the electron transfer process during photocatalysis [106].

For improved photocatalytic H_2_ evolution, a widely studied example of anchoring a MoS_2_ co-catalyst on graphene has been reported [107,108]. Xiang et al., synthesized a TiO_2_/MoS_2_/graphene hybrid photocatalyst for hydrogen production [18]. The hybrid photocatalyst showed significant enhancement of photocatalytic H_2_ generation under UV illumination, with an apparent quantum efficiency of 9.7% at 365 nm. The improved activity is described in terms of synergetic effects between MoS_2_ and the conductive graphene co-catalysts and TiO_2_ leading to outstanding photocatalytic hydrogen evolution activity. These authors have proposed a mechanism for the significant boost of photocatalytic H_2_ generation. They reported that this enhancement is due to the transfer of photogenerated electrons from the CB of TiO_2_ nanoparticles to the CB of MoS_2_ nanosheets through highly conductive graphene sheets (Figure 7), where H^+^ ions are adsorbed at an active site of MoS_2_. Apart from graphene, other highly conductive materials such as metal sulfides and phosphides can also be used as interfacial electron transfer sources to enhance photocatalytic hydrogen evolution. Lu and coworkers synthesized g-C_3_N_4_, Ni_2_P, and MoS_2_ heterojunctions by hydrothermal and ultrasonic methods [109]. The hydrogen production rate of the g-C_3_N_4_-1%Ni_2_P-1.5%MoS_2_ composite was about 532.41 μmol·h^−1^·g^−1^ under visible light, which is 5.15- and 2.47-fold higher than those of g-C_3_N_4_-1%Ni_2_P and g-C_3_N_4_-1.5%MoS_2_, respectively. The Ni_2_P co-catalyst could be acting as an interface electron bridge between g-C_3_N_4_ and MoS_2_ nanosheets. It provides interfacial electron transfer channels in g-C_3_N_4_/MoS_2_ heterostructure composites and prevents the rapid recombination process of photogenerated charge carriers.

Finally, we summarize some heterojunction composites with semiconducting and MoS_2_ materials in which the MoS_2_ nanomaterial acts as a co-catalyst for enhanced photocatalytic hydrogen production. Table 1 and Figure 8 show different strategies used for various types of catalysts combined with a MoS_2_ co-catalyst to form heterojunction composites for enhanced photocatalytic hydrogen production.

**Table 1 molecules-27-03289-t001:** Summary of MoS_2_ usage as a co-catalyst for various materials to form heterostructures for photocatalytic hydrogen generation.

Catalyst	Synthesis Method	Light Source	Photocatalytic Activity	No. of Cycles	Total Times of Cycles (h)	Ref.
MoS_2_ nanoparticles/TiO_2_ nanoparticles	Mechanochemistry	300 W Xe lamp (λ = 250–380 nm)	150.7 μmol·h^−1^·g^−1^	3	18	[76]
TiO_2_ nanofibers @MoS_2_ nanosheets	Hydrothermal	300 W xenon lamp λ > 320 nm or λ > 420 nm	1.68 mmol·h^−1^·g^−1^ 0.49 mmol·h^−1^·g^−1^	6	30	[19]
Flower-like MoS_2_@TiO_2_ nanohybrids	Metal organic framework-derived	300 W Xe lamp (λ ≥ 420 nm)	10046 µmol ·h^−1^·g^−1^	3	10	[77]
MoS_2_ nanosheets/TiO_2_ nanotubes	Hydrothermal process	300 W Xe-lamp (λ ≥ 420 nm)	143.32 μmol·h^−1^·g^−1^	4	14	[110]
MoS_2_ nanosheets/g-C_3_N_4_ nanosheets	Solvothermal method	300 W Xe-lamp (λ > 420 nm	1155 μmol·h^−1^·g^−1^	3	12	[87]
S-doped C_3_N_4_ nanosheets/MoS_2_ nanosheets	One-step solid-state strategy	Visible LED lamp	702.53 μmol·h^−1^·g^−1^	3	16	[88]
Amorphous MoS_x_ nanoparticles/g-C_3_N_4_ nanosheets	Adsorption in situ transformation method	Low-power LEDs (3W, 420 nm)	273.1 μmol·h^−1^·g^−1^	4	12	[95]
g-C_3_N_4_/NCDS/MoS_2_	Thermal polymerization and solvothermal approach	300 W Xe lamp (λ ≥ 420 nm)	212.41 μmol·h^−1^·g^−1^	4	16	[111]
ZnS/MoS_2_ particles	One-pot solvothermal	Hg pen-lamp (254 nm), (4.4 mW/cm^2^)	606 μmol·h^−1^·g^−1^	-	-	[101]
MoS_2_ clusters/CdS nanorod	Solvothermal method	300 W Xe lamp (λ ≥ 420 nm)	12.38 mmol·h^−1^·g^−1^	4		[112]
MoS_2_/ZnIn_2_S_4_ microspheres	Impregnation method	300 W Xe-lamp (λ > 420 nm)	3.06 mmol·h^−1^·g^−1^	3	15	[113]
MoS_2_ nanosheets/ZnIn_2_S_4_ microspheres	In situ photo-assisted deposition	300 W Xe-lamp (λ > 420 nm)	8.047 mmol·h^−1^·g^−1^	-	-	[114]
MoS_2_ nanoflake-Mn_0_._2_Cd_0_._8_S nanorod/MnS nanoparticle	One-pot solvothermal	300 W Xe lamp (λ ≥ 420 nm)	995 μmol·h^−1^	5	20	[115]

## 6. Conclusions and Outlook

In summary, we highlighted the significance of MoS_2_ as a co-catalyst to improve hydrogen evolution. A comprehensive analysis of the literature led us to conclude that MoS_2_ is a good co-catalyst for other semiconducting materials such as TiO_2_, C_3_N_4_, CdS, ZnS, etc., which form heterostructure nanocomposites and consequently boost the photocatalytic hydrogen generation ability. However, there are still some critical issues that must be resolved, such as the downsizing of MoS_2_ nanosheets for appropriate band gap alignment and the high density of catalytic active sites. These issues can be solved by reducing the size of MoS_2_ to quantum dots or the molecular level, which will certainly enhance the catalytic active sites. The photoexcited electron transfers between photocatalysts and the MoS_2_ co-catalyst play an important role during photocatalytic hydrogen generation. The electron transfer mechanism at the interface of a semiconductor photocatalyst and a MoS_2_ co-catalyst is yet to be fully investigated and completely understood. It is important to conduct theoretical studies such as density functional theory (DFT) simulations and apply in situ testing methods to understand electron transfer paths. Although MoS_2_ nanosheets as a co-catalyst are a promising candidate for photocatalytic hydrogen production, all the challenges require further efforts and study.

## Figures and Tables

**Figure 1 molecules-27-03289-f001:**
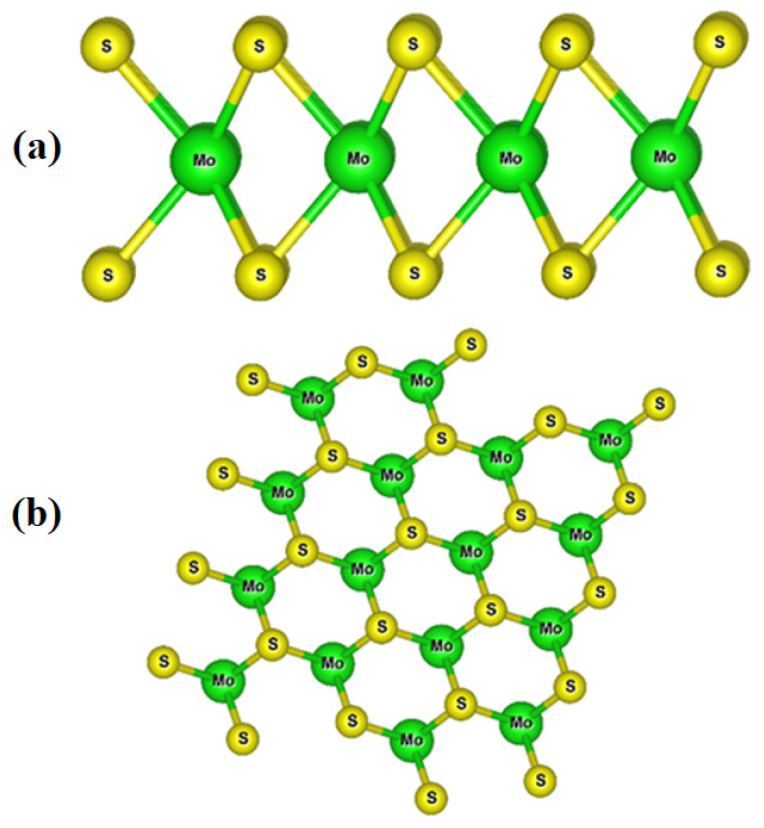
(**a**) Side and (**b**) top views of MoS_2_ single layer.

**Figure 2 molecules-27-03289-f002:**
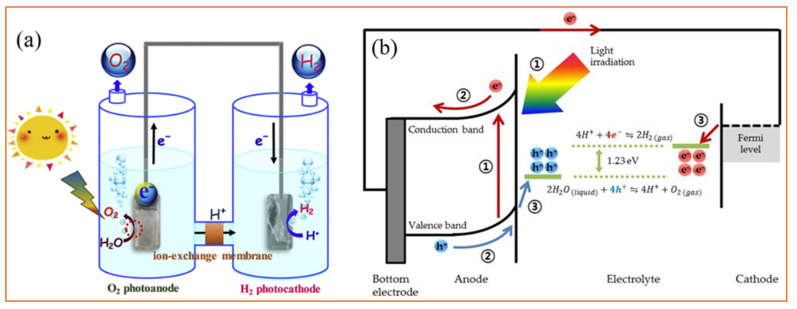
(**a**) Schematic device illustration of photoelectrochemical water splitting. Reprinted with permission from Ref. [38] (Copyright 2019 Elsevier). (**b**) Schematic representation of the photoelectrochemical water splitting process in a common PEC water splitting system consisting of a photoanode and a metal counterpart. Reprinted from Ref. [39].

**Figure 3 molecules-27-03289-f003:**
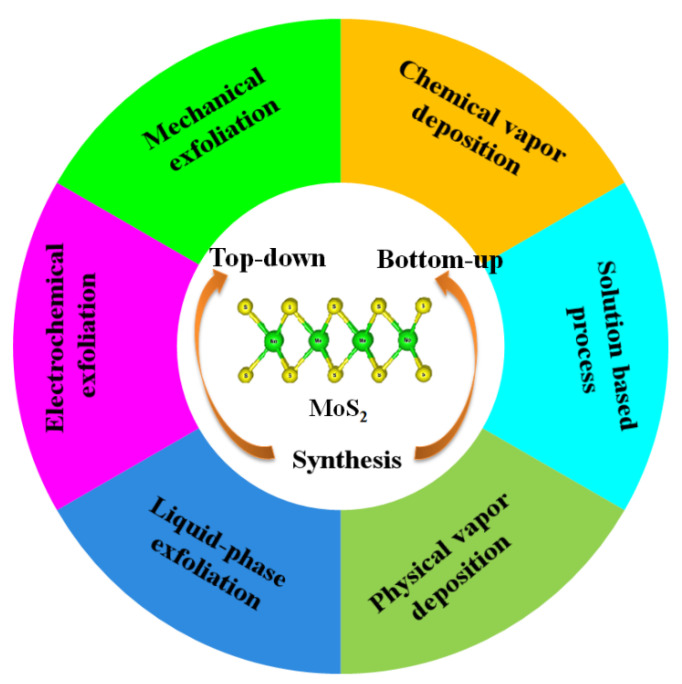
Various synthetic methods for MoS_2_ preparation.

**Figure 4 molecules-27-03289-f004:**
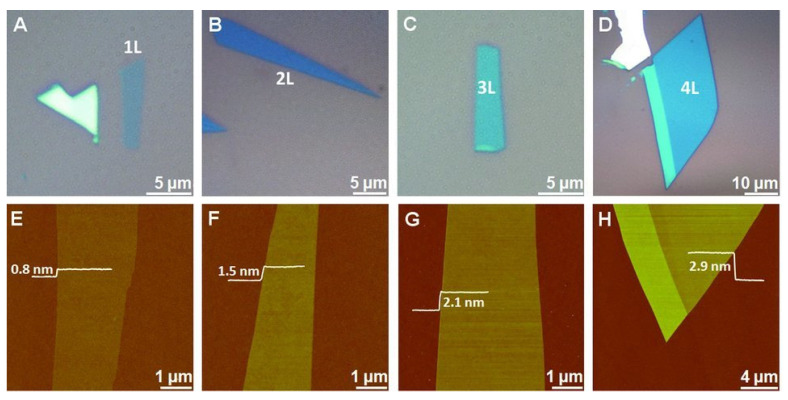
Mechanically exfoliated single- and multilayer MoS_2_ nanosheets on Si/SiO_2_. (**A**–**D**) Optical microscope and (**E**–**H**) AFM images of MoS_2_ nanosheets. The single MoS_2_ sheet thickness is 0.8 nm (**E**), while the thickness of two (**F**), three (**G**), and four (**H**) layers of MoS_2_ nanosheets is 1.5, 2.1, and 2.9 nm, respectively. Reprinted with permission from Ref. [41]. Copyright 2012 Wiley-VCH Verlag GmbH & Co.

**Figure 5 molecules-27-03289-f005:**
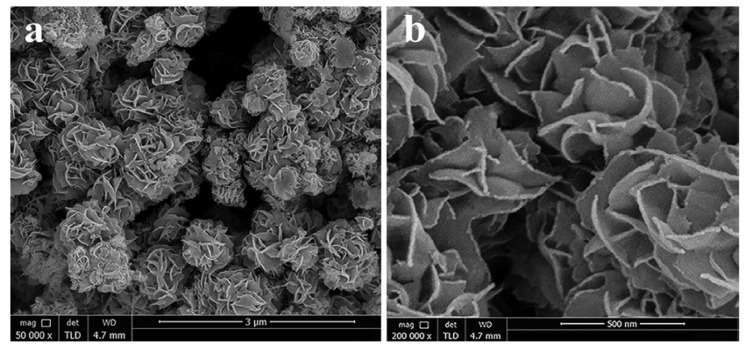
(**a**,**b**) SEM images of MoS_2_@TiO_2_ composites. Reprinted with permission from Ref. [77]. Copyright 2016 American Chemical Society.

**Figure 6 molecules-27-03289-f006:**
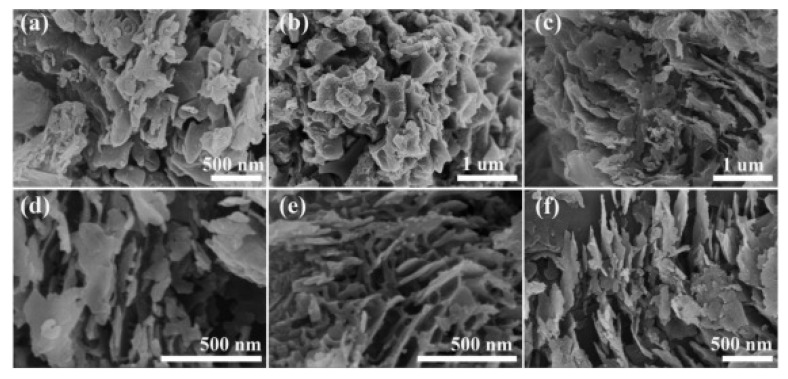
SEM images of (**a**) SC_3_N_2_, (**b**) MoS_2_ and ultrathin array-like nanosheet, (**c**) MoS_2_/SC_3_N_4_-0.5%, (**d**) MoS_2_/SC_3_N_4_-1.5%, (**e**), MoS_2_/SC_3_N_4_-2.5%, and (**f**) MoS_2_/SC_3_N_4_-5.0%. Reprinted with permission from Ref. [88]. Copyright 2021 Elsevier.

**Figure 7 molecules-27-03289-f007:**
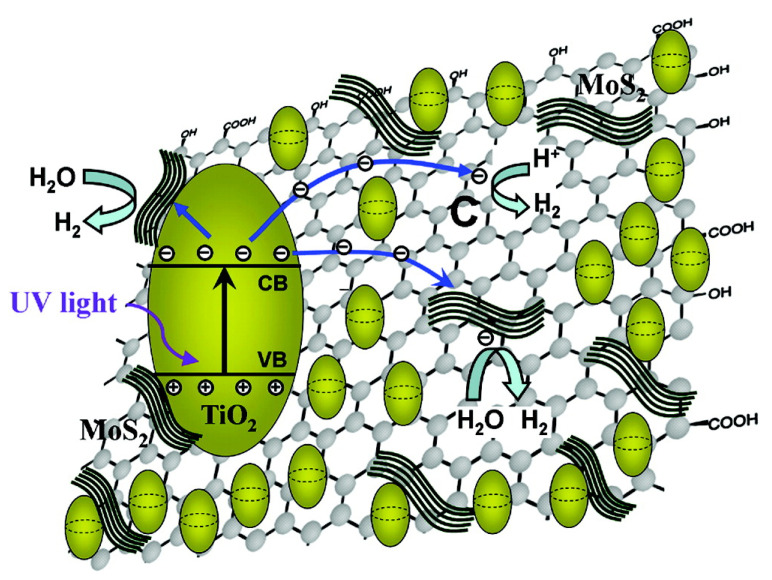
Schematic illustration of the charge transfer and proposed mechanism of electron transfer in TiO_2_/MoS_2_/graphene composites. Reprinted with permission from Ref. [18]. Copyright 2012 American Chemical Society.

**Figure 8 molecules-27-03289-f008:**
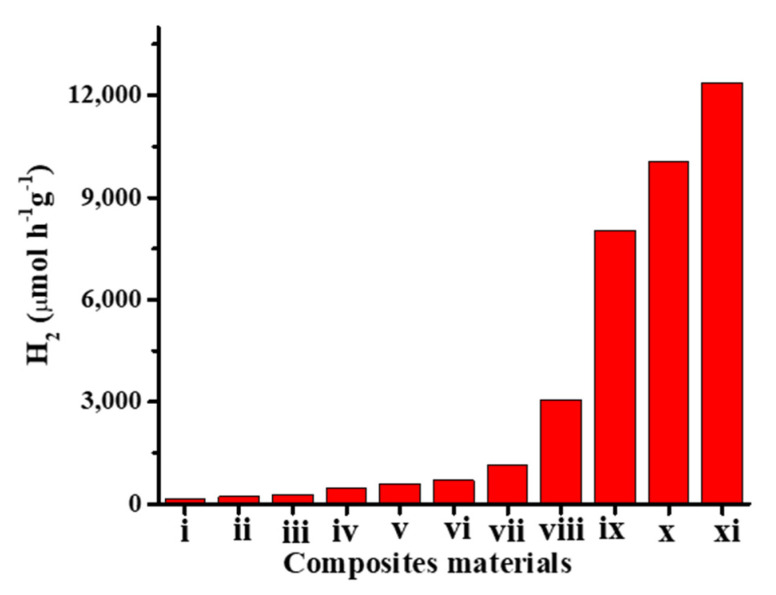
Photocatalytic hydrogen production of heterojunction materials using MoS_2_ as a co-catalyst. (i) MoS_2_ nanosheets/TiO_2_ nanotubes [110]. (ii) g-C_3_N_4_/NCDS/MoS_2_ [111]. (iii) Amorphous MoS_x_ nanoparticles/g-C_3_N_4_ nanosheets [95]. (iv) TiO_2_ nanofibers/@MoS_2_ nanosheets [19]. (v) ZnS/MoS_2_ particles [101]. (vi) S-doped C_3_N_4_ nanosheets/MoS_2_ nanosheets [88]. (vii) MoS_2_ nanosheets/g-C_3_N_4_ nanosheets [87]. (viii) MoS_2_/ZnIn_2_S_4_ microspheres [113]. (ix) MoS_2_ nanosheets/ZnIn_2_S_4_ microspheres [114]. (x) Flower-like MoS_2_@TiO_2_ nanohybrids [77]. (xi) MoS_2_ clusters/CdS nanorod [112].

## Data Availability

No supporting data is reported.

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
