# Peer review of "MoS2 as a Co-Catalyst for Photocatalytic Hydrogen Production: A Mini Review"

_molecules, 2022, doi:10.3390/molecules27103289_

Round 1
Reviewer 1 Report
The author summarized the basic aspects and synthetic methods of MoS2 nanosheets. Besides, they discussed the formation of MoS2 heterostructure with other semiconductors and cocatalysts, for the enhanced photocatalytic hydrogen generation. The authors did a great job reviewing this challenging topic. I would be very happy to see this review in press soon.
However I have some remarks, it would be great if the authors consider them before publication:
- In abstract I think you need to add some data from the review.
- At the of introduction please add objective of the review.
- I recommend the authors to include the publication from
Yan Zhang, Junfen Wan, Chunjuan Zhang & Xuejun Cao (2022). MoS2 and Fe2O3 co-modify g-C3N4 to improve the performance of photocatalytic hydrogen production. Scientific Reports volume 12, Article number: 3261.
- If possible add table summerize Application of MoS2 as a cocatalyst in photocatalysis for hydrogen production.
- you need to seperate section by lines, Also you need to justify the titles of figures.
- Please add update by 2022 References
Author Response
Response (s) to Reviewer # 1 Comments
Comment 1: In abstract I think you need to add some data from the review.
Response: Thank you very much for your suggestion. The abstract of the manuscript is revised and some sentences are added to it.
“Abstract: Molybdenum disulfide (MoS2) with a two-dimensional 2D structure, has attracted huge research interest due to its unique electrical, optical, and physicochemical properties. MoS2 has been used as cocatalysts for the synthesis of novel heterojunction composites with enhanced photocatalytic hydrogen production under solar light irradiation. In this review, we briefly highlight atomic-scale structure of MoS2 nanosheets. The top-down and bottom-up synthetic methods of MoS2 nanosheets are described. Besides, we discussed the formation of MoS2 heterostructure with titanium dioxide (TiO2), graphitic carbon nitride (g-C3N4), other semiconductor and cocatalysts, for the enhanced photocatalytic hydrogen generation. This review addresses the challenges and future perspectives for enhancing solar hydrogen production performance in the heterojunction materials using MoS2 as a cocatalyst.”
Comment 2. At the of introduction please add objective of the review.
Response: Thank you very much. In the introduction part, we added some seme sentences to describe the objective of the manuscript.
“In this review, we briefly introduce basic aspects and synthetic methods of MoS2 nanosheets. Different type of MoS2-based of heterojunction composites are also discussed. The role of MoS2 nanomaterials as co-catalyst in the heterojunction composites for enhanced HER performance are addressed. Some important issues are highlighted and useful opinion are discussed to further improve photocatalytic hydrogen production using MoS2 as co-catalyst.”
Comment 3. I recommend the authors to include the publication from
Yan Zhang, Junfen Wan, Chunjuan Zhang & Xuejun Cao (2022). MoS2 and Fe2O3 co-modify g-C3N4 to improve the performance of photocatalytic hydrogen production. Scientific Reports volume 12, Article number: 3261.
Response: Thanks. The above paper is properly cited in the revised manuscript.
“Zhang et al. reported MoS2/Fe2O3/g-C3N4 ternary composite photocatalyst under hydrothermal conditions for hydrogen production [90]. The obtained ternary composite shows the hydrogen production rate about 5 times higher as compared to g-C3N4.”
Comment 4. If possible, add table summerize Application of MoS2 as a cocatalyst in photocatalysis for hydrogen production.
Response: Thanks. The Table 1 summarized the application of MoS2 as a cocatalyst in photocatalysis for hydrogen production.
Table 1. Summary of MoS2 as a cocatalyst for various materials to form heterostructures for photocatalytic hydrogen generation.
|
Catalyst |
Synthesis method |
Light source |
Photocatalytic activity |
No: cycles |
Total times of cycles (h) |
Ref.
|
|
MoS2 nanoparticles/TiO2 nanoparticles |
mechanochemistry |
300 W Xe lamp (λ= 250–380 nm) |
150.7 μmol·h-1·g-1 |
3 |
18 |
76 |
|
TiO2 nanofibers @MoS2 nanosheets |
hydrothermal |
300 W xenon lamp λ > 320 nm or λ > 420 nm |
1.68 mmol·h-1·g-1 0.49 mmol·h-1·g-1 |
6 |
30 |
19 |
|
Flower-like MoS2@TiO2 nanohybrids |
metal-organic framework-derived |
300 W Xe lamp (λ ≥ 420 nm) |
10046 µmol ·h-1·g-1 |
3 |
10 |
77 |
|
MoS2 nanosheets /TiO2 nanotubes |
hydrothermal process |
300 W Xe-lamp (λ ≥ 420 nm) |
143.32 μmol·h-1·g-1 |
4 |
14 |
110 |
|
MoS2 nanosheets /g-C3N4 nanosheets |
solvent-thermal method |
300 W Xe-lamp (λ > 420 nm |
1155 μmol·h-1·g-1 |
3 |
12 |
87 |
|
S-doped C3N4 nanosheets /MoS2 nanosheets |
One step solid-state strategy |
Visible LED lamp |
702.53 μmol·h-1·g-1 |
3 |
16 |
88 |
|
Amorphous MoSx nanoparticles /g-C3N4 nanosheets |
adsorption-in situ transformation method |
low-power LEDs (3W, 420 nm) |
273.1 μmol·h-1·g-1 |
4 |
12 |
95 |
|
g-C3N4/NCDS/ MoS2 |
thermal polymerization and solvothermal approach |
300 W Xe lamp (λ ≥ 420 nm) |
212.41 μmol·h-1·g-1 |
4 |
16 |
111 |
|
ZnS/MoS2 particles |
one-pot solvothermal |
Hg pen-lamp (254 nm, (4.4 mW/cm2 ) |
606 μmol·h-1·g-1 |
- |
- |
101 |
|
MoS2 clusters /CdS nanorod |
solvothermal method |
300 W Xe lamp (λ ≥ 420 nm) |
12.38 mmol·h-1·g-1 |
4 |
|
112 |
|
MoS2/ZnIn2S4 microspheres |
impregnation method |
300 W Xe-lamp (λ > 420 nm |
3.06 mmol·h-1·g-1 |
3 |
15 |
113 |
|
MoS2 nnosheets /ZnIn2S4 microspheres |
in-situ photo-assisted deposition |
300 W Xe-lamp (λ > 420 nm |
8.047 mmol·h-1·g-1 |
- |
- |
114 |
|
MoS2 nanoflake-Mn0.2Cd0.8S nanorod /MnS nanoparticle |
one-pot solvothermal |
300 W Xe lamp (λ ≥ 420 nm) |
995 μmol·h-1 |
5 |
20 |
115 |
Comment 5. you need to seperate section by lines, Also you need to justify the titles of figures.
Response: Thank you very much. We separated each section by lines. The caption of each Figure justifies in the text of manuscript.
Comment. Please add update by 2022 References
Response: References from 2022 were added and cited in the revised manuscript.
“Recently, Li et al. reported the in-situ synthesis of g-CN/MoS2 composite [89]. The composite exhibits an enhanced photocatalytic hydrogen production compared to pristine g-CN under visible light irradiation. The rodlike MoS2 play an important role as cocatalyst in the g-CN/MoS2 composite for enhancement of hydrogen production rate.”
“Zhang et al. reported MoS2/Fe2O3/g-C3N4 ternary composite photocatalyst under hydrothermal conditions for hydrogen production [90]. The obtained ternary composite shows the hydrogen production rate about 5 times higher as compared to g-C3N4.”
“Patriarchea and coworker synthesized CdS nanoparticles by a polymer-templated oxidative aggregation and subsequently MoS2 nanosheets were deposited on it via wet-chemical method [100]. The obtained optimiedMoS2/CdS catalyst shows good hydrogen production rate about 0.4 mmol h-1 under visible light compared to CdS catalyst. The enhanced hydrogen generation is due to the presence of MoS2 cocatalyst.”
Reviewer 2 Report
1) Abstract: could be improved especially for last sentence: we suggest that this review ……what is the basis for this suggestion?
2) There are many abbreviations used throughout the manuscript. A detailed list of abbreviations could be tabulated at the start of the manuscript.
3)Font size for each the title of Figures, eg. Figure 2.(a) ….. is too big & recommend to be one to two size smaller than main text.
4) Chapter 3: this chapter is too brief and not informative. Can it even quantify as a chapter by itself?
5) Figure 2: Size and resolution is blurry for 2(b).
6) For each sub-chapter in Chapter 4 and 5, many examples from literatures were mentioned and quoted. However, there is no conclusion/summary to summarize each sub-chapter. There seems to be a breakdown in flow and storyline continuity between the sub-chapters.
7) Size for Figure 7 is really huge. The authors ensure the size and resolution for all figure to be consistent.
8) There is no mention on what is the significance for Table 1 and Figure 8. There ought to be some description/information to summarise table 1 and Figure 8.
9) References: Formatting and styles to be check carefully for consistencies.

Author Response
Response (s) to Reviewer # 2 Comments
Comment 1. Abstract: could be improved especially for last sentence: we suggest that this review ……what is the basis for this suggestion?
Response: Thank you very much. We have improved the abstract of the manuscript according to your suggestion.
“Abstract: Molybdenum disulfide (MoS2) with a two-dimensional 2D structure, has attracted huge research interest due to its unique electrical, optical, and physicochemical properties. MoS2 has been used as cocatalysts for the synthesis of novel heterojunction composites with enhanced photocatalytic hydrogen production under solar light irradiation. In this review, we briefly highlight atomic-scale structure of MoS2 nanosheets. The top-down and bottom-up synthetic methods of MoS2 nanosheets are described. Besides, we discussed the formation of MoS2 heterostructure with titanium dioxide (TiO2), graphitic carbon nitride (g-C3N4), other semiconductor and cocatalysts, for the enhanced photocatalytic hydrogen generation. This review addresses the challenges and future perspectives for enhancing solar hydrogen production performance in the heterojunction materials using MoS2 as a cocatalyst.”
Comment 2. There are many abbreviations used throughout the manuscript. A detailed list of abbreviations could be tabulated at the start of the manuscript.
Response: Thank you very much for your suggestion. All the words are abbreviated, when it appeared for the first time. We read many papers of Journal “Molecules” and no separate abbreviations list were present in it.
Comment 3. Font size for each the title of Figures, eg. Figure 2.(a) ….. is too big & recommend to be one to two size smaller than main text.
Response: Thank you very much. The font size of each Figure caption is adjusted two size smaller than main text.
Comment 4. Chapter 3: This chapter is too brief and not informative. Can it even quantify as a chapter by itself?
Response: Thank you very much. This chapter just the brief introduction photochemical hydrogen evolution reaction in the mini review. However, we added few sentences for more understanding and information to Chapter 3.
“As mentioned above, Fujishima and Honda reported the photo-induced water splitting on TiO2 electrodes. The hydrogen can also be directly produced from photochemical water splitting. Usually, a photoelectrolytic cell is designed to carry out a photochemical water splitting process. A typical photoelectrolytic cell for water splitting is shown in Figure 2a [38]. Using light sources, the photocatalytic water splitting take place in several steps. The absorption of light by catalyst on electrode. The generation of charges and followed by excitation of electrons in the valence band. The separation of charge as well as transport of charge carriers. The oxidation of water and generation of hydrogen darning water splitting are occurred at respective electrodes. The pure overall water splitting comprises of two half-reactions to generate hydrogen and oxygen molecules Figure 2b [39]. The water oxidation at the anode to produced oxygen, meanwhile H+ ions are reduced on cathode into hydrogen gas. For more details photocatalytic water splitting read review of Jeong et al. [39].”
Comment 5. Figure 2: Size and resolution is blurry for 2(b).
Response: Thanks. we tried to improve the quality of Figure 2(b).
Figure 2. (a) Schematic device illustration of Photoelectrochemical water splitting Reprinted with permission from ref. [38] Elsevier 2019, (b) and Schematic representation of the photoelectrochemical water splitting process in a common PEC water splitting system consisting of a photoanode and a metal counterpart. Reprinted with Permission from ref. [39] MDPI 2018.
Comment 6. For each sub-chapter in Chapter 4 and 5, many examples from literatures were mentioned and quoted. However, there is no conclusion/summary to summarize each sub-chapter. There seems to be a breakdown in flow and storyline continuity between the sub-chapters.
Response: Thank you very much for your suggestion. For each sub-chapter in Chapter 4 and 5, We added some sentence to drawn some conclusion from them.
“In the top-down approaches from single to multilayer MoS2 nanosheets is prepared, which used to study some fundamental properties of MoS2 nanosheets.”
“The MoS2 nanomaterials prepared through different bottom-up approaches have various sizes, shapes, morphologies, and thickness and can be used for many applications.”
“The TiO2 nanomaterials combined with MoS2 cocatalyst can enhance the hydrogen production rates up to several time.”
“Similar to TiO2/MoS2 photocatalysts, MoS2/g-C3N4 heterojunction composites can be improved the hydrogen production.”
“For these examples, we can conclude the MoS2 is efficient cocatalyst for the CdS, ZnS, and MAPbI3, etc. catalysts to produce hydrogen.”
Comment 7. Size for Figure 7 is really huge. The authors ensure the size and resolution for all figure to be consistent.
Response: Thanks. The size of Figure 7 was reduced. We tried our best to maintain the consistency in all Figures.
Comment 9. There is no mention on what is the significance for Table 1 and Figure 8. There ought to be some description/information to summarise table 1 and Figure 8.
Response: Thanks. We revised the text to described the significance of Table 1 and Figure 8.
At last, we summarized some heterojunction composites between semiconducting and MoS2 materials in which the MoS2 nanomaterials acts as cocatalyst for enhanced photocatalytic hydrogen production. Table1 and Figure 8 show different strategies used for various types of catalysts combined with MoS2 nanomaterials cocatalyst to formed heterojunction composites for enhanced photocatalytic hydrogen production.
Comment 10. References: Formatting and styles to be check carefully for consistencies.
Response: Thanks. We tried our best to correct the format and style of all references.

Reviewer 3 Report
The MoS2 materials as a catalyst are widely studied and already few reviews had been published on photocatalytic applications such as degradation & H2 production. However, H2 production via photocatalytic over MoS2-related catalysts is also studied extensively, but recent advances in MoS2 development are still not comprehensively reported. In this work, the review was not written extensively. There are still a few important things that are missing such as reporting and summarising the MoS2 in combination with catalysts or scaffolds are not reported and also various H2 sources.

Author Response
Response (s) to Reviewer # 3 Comments
Comments. The MoS2 materials as a catalyst are widely studied and already few reviews had been published on photocatalytic applications such as degradation & H2 production. However, H2 production via photocatalytic over MoS2-related catalysts is also studied extensively, but recent advances in MoS2 development are still not comprehensively reported. In this work, the review was not written extensively. There are still a few important things that are missing such as reporting and summarising the MoS2 in combination with catalysts or scaffolds are not reported and also various H2 sources.
Responses: Thank you very much for your kind suggestion. We have added some recent examples to this mini review. We summarized the various catalyst combine with MoS2, synthesis method, photocatalytic activity and stability etc. in Table 1.
“Recently, Li et al. reported the in-situ synthesis of g-CN/MoS2 composite [89]. The composite exhibits an enhanced photocatalytic hydrogen production compared to pristine g-CN under visible light irradiation. The rodlike MoS2 play an important role as cocatalyst in the g-CN/MoS2 composite for enhancement of hydrogen production rate.”
“Zhang et al. reported MoS2/Fe2O3/g-C3N4 ternary composite photocatalyst under hydrothermal conditions for hydrogen production [90]. The obtained ternary composite shows the hydrogen production rate about 5 times higher as compared to g-C3N4.”
“Patriarchea and coworker synthesized CdS nanoparticles by a polymer-templated oxidative aggregation and subsequently MoS2 nanosheets were deposited on it via wet-chemical method [100]. The obtained optimiedMoS2/CdS catalyst shows good hydrogen production rate about 0.4 mmol h-1 under visible light compared to CdS catalyst. The enhanced hydrogen generation is due to the presence of MoS2 cocatalyst.”
Table 1. Summary of MoS2 as a cocatalyst for various materials to form heterostructures for photocatalytic hydrogen generation.
|
Catalyst |
Synthesis method |
Light source |
Photocatalytic activity |
No: cycles |
Total times of cycles (h) |
Ref.
|
|
MoS2 nanoparticles/TiO2 nanoparticles |
mechanochemistry |
300 W Xe lamp (λ= 250–380 nm) |
150.7 μmol·h-1·g-1 |
3 |
18 |
76 |
|
TiO2 nanofibers @MoS2 nanosheets |
hydrothermal |
300 W xenon lamp λ > 320 nm or λ > 420 nm |
1.68 mmol·h-1·g-1 0.49 mmol·h-1·g-1 |
6 |
30 |
19 |
|
Flower-like MoS2@TiO2 nanohybrids |
metal-organic framework-derived |
300 W Xe lamp (λ ≥ 420 nm) |
10046 µmol ·h-1·g-1 |
3 |
10 |
77 |
|
MoS2 nanosheets /TiO2 nanotubes |
hydrothermal process |
300 W Xe-lamp (λ ≥ 420 nm) |
143.32 μmol·h-1·g-1 |
4 |
14 |
110 |
|
MoS2 nanosheets /g-C3N4 nanosheets |
solvent-thermal method |
300 W Xe-lamp (λ > 420 nm |
1155 μmol·h-1·g-1 |
3 |
12 |
87 |
|
S-doped C3N4 nanosheets /MoS2 nanosheets |
One step solid-state strategy |
Visible LED lamp |
702.53 μmol·h-1·g-1 |
3 |
16 |
88 |
|
Amorphous MoSx nanoparticles /g-C3N4 nanosheets |
adsorption-in situ transformation method |
low-power LEDs (3W, 420 nm) |
273.1 μmol·h-1·g-1 |
4 |
12 |
95 |
|
g-C3N4/NCDS/ MoS2 |
thermal polymerization and solvothermal approach |
300 W Xe lamp (λ ≥ 420 nm) |
212.41 μmol·h-1·g-1 |
4 |
16 |
111 |
|
ZnS/MoS2 particles |
one-pot solvothermal |
Hg pen-lamp (254 nm, (4.4 mW/cm2 ) |
606 μmol·h-1·g-1 |
- |
- |
101 |
|
MoS2 clusters /CdS nanorod |
solvothermal method |
300 W Xe lamp (λ ≥ 420 nm) |
12.38 mmol·h-1·g-1 |
4 |
|
112 |
|
MoS2/ZnIn2S4 microspheres |
impregnation method |
300 W Xe-lamp (λ > 420 nm |
3.06 mmol·h-1·g-1 |
3 |
15 |
113 |
|
MoS2 nnosheets /ZnIn2S4 microspheres |
in-situ photo-assisted deposition |
300 W Xe-lamp (λ > 420 nm |
8.047 mmol·h-1·g-1 |
- |
- |
114 |
|
MoS2 nanoflake-Mn0.2Cd0.8S nanorod /MnS nanoparticle |
one-pot solvothermal |
300 W Xe lamp (λ ≥ 420 nm) |
995 μmol·h-1 |
5 |
20 |
115 |
